# Respectful family planning service provision in Sidama zone, Southern Ethiopia

**Melese Siyoum**[1]*, **Ayalew Astatkie**[2], **Zelalem Tenaw**[1], **Abebaw Abeje**[1], **Teshome Melese**[1]

1 Department of Midwifery, College of Medicine and Health Sciences, Hawassa University, Hawassa, Ethiopia, 2 School of Public Health, College of Medicine and Health Sciences, Hawassa University, Hawassa, Ethiopia

* melesesiyoum755@gmail.com

## Abstract

### Introduction

Disrespect and abusive care is a violation of women's basic human rights and it is serious global problem that needs urgent intervention. Poor quality client-provider interaction is commonly reported from family planning programmes. In Ethiopia, disrespect and abusive care is very common (21–78%) across health facilities.

### Objective

To assess the status of respectful family planning service (client-provider interaction) in Sidama zone, south Ethiopia.

### Methodology

Health facility-based cross-sectional study was conducted from June to August 2018. Data were collected from 920 family planning clients recruited from 40 randomly selected health facilities. The Mother on Respect index (MORi) questionnaire was used to collect the data through client exit interview. Partial proportional odds ordinal regression was employed to identify determinants of respectful family planning service.

### Result

Among family planning clients, the level of respectful family planning service was found to be zero (0%) in the very low respect category, 75(18.5%) in the low respect category, 382 (41.52%) in moderate respect category and 463(50.33%) in high respect category. Being a short acting method client (AOR = 0.30, 95%CI [0.12, 0.72]), being an uneducated client (AOR = 0.39, 95%CI [0.25, 0.61]) or a client with elementary education (AOR = 0.41, 95%CI [0.23, 0.73]), client's poverty (AOR = 0.75, 95%CI [0.56, 0.99]), and long waiting time (AOR = 0.46, 95%CI [0.30, 0.69])significantly reduced the odds of moderate and high respect compared to low respect. Conversely, preference of male service providers, service providers' work satisfaction and health workers' prior training on respectful care significantly increased the odds of moderate and high respect.

**Data Availability Statement:** The data underlying the results of this study are provided in the article and its supporting information.

**Funding:** This study was supported by Center for International Reproductive Health Training

(CIRHT), Ethiopia. The funders had no role in study design, data collection and analysis, decision to publish, or preparation of the manuscript.

**Competing interests:** The authors have declared that there is no conflict of interest.

## Conclusion

Considering the current strategy of zero tolerance for disrespect and abuse in Ethiopia, the level of respectful care in this study is sub-optimal. Short term training for service providers on respectful care seems valuable to enhance the level of respectful care for family planning clients irrespective of their socioeconomic background.

## Back ground

Respectful maternity care (RMC) is an individual (client) centered approach, which is based on principles of ethics and respect for human rights and women's needs and preferences [1]. There are many evidence based definitions of disrespectful maternity care [2–8]. The most commonly used definition is proposed by Bowser and Hill (2010), which includes physical abuse, non-consented care, non-confidential care, non-dignified care, discrimination, abandonment of care and detention in health facility [1, 5, 9–13]. Disrespect and abuse of service seeking women is an urgent problem that needs multisector response including health care research, quality and education, human rights and civil rights advocacy throughout the entire world [3].

A growing body of evidence shows that disrespect and abuse during maternity care is becoming an increasing problem worldwide [1, 3, 7, 10, 14]. A cross-sectional study conducted in Nigeria showed that 98% of childbearing mothers had faced at least one category of disrespect [15]. A systematic review conducted in the same country revealed that disrespect during child birth is common and the most frequent category of disrespect was non-dignified care (11.3% to 70.8% across the health facilities [16]. In Tanzania, 70% of respondents were disrespected where non-consented care was reported in 50% of participants [12].

A study done in Addis Ababa, Ethiopia showed that 78% of women who gave birth at health facilities experienced one or more type of disrespect. The right for information such as self-introduction and consent was violated in 94% of the cases [17]. It is common to see disrespect and abusive care at the point of client-provider interaction [2, 4, 11, 14, 18]. In addition to the high prevalence of disrespect and abusive care worldwide, it is neglected and even considered normal in many areas [5, 10, 16, 19]. For instance among 78.6% of participants who were disrespected based on observational checklist in Addis Ababa during child birth, only 22% of them considered it a disrespectful care [17].

Lack of RMC constitutes a barrier to the use of health service and it affects the basic human dignity and human rights. Disrespectful and abusive behavior tends to be higher among women identified as having low socioeconomic status and lower educational attainment. Disrespectful care is also higher when there are weak health care systems, poor managerial systems, provider demotivation, lack of equipment and supplies, and weak or non-existing legal system [16, 19–22].

Though RMC continues to be a problem, there is evidence that interventions such as health care provider training and community mobilization have a positive impact on the promotion of RMC [4, 15, 18, 19]. To overcome the problem of disrespectful and abusive care in Ethiopia, L10K and the Ethiopian Midwives Association have been providing training for higher institution instructors and obstetric care providers who are working in the labor and delivery ward [14, 23]. The Ministry of Health has also started to provide training for all health care workers since March 2017. However, family planning units where respectful care is highly needed as women need correct and accurate information, privacy and confidentiality, respect for their choice of method, dignity and freedom from physical abuse, are still neglected. In Ethiopia the contraceptive prevalence rate is low (36%) for married women. There is also low uptake of

long acting methods, high discontinuation rate of long acting methods and poor counseling services [24], all of which could be related to lack of respectful care, yet there is lack of evidence on the level of respectful care for family planning service users in Ethiopia. While qualitative researches show poor client-provider interaction during family planning provision, there is lack of evidence on the prevalence of disrespect and abuse [25]. Therefore, this study aimed to assess the level of respectful family planning services and its determinants in Sidama Zone, southern Ethiopia. Unlike previous qualitative studies, disrespect and abusive care during family planning service are quantitatively measured.

## Methods and materials

### Study design and setting

This study was a health facility-based cross sectional study conducted in Sidama zone, Southern Ethiopia from 29 June to 20 July 2018. Hawassa, the capital city of the Southern Nations, Nationalities and Peoples Region (SNNPR) and Sidama Zone, is 273 km south of Addis Ababa, the capital of Ethiopia. The zone is situated in the southern region of the country and has 19 districts and 3 town administrations. According to a report from Sidama Zone Health Department, the total population of Sidama zone is estimated to be 5,499,683 of which 719,937 are women of reproductive age. There are three governmental hospitals, 130 health centers and 522 health posts providing family planning services. The overall contraceptive coverage of SNNPR is 39.9% [26]; and the coverage for long acting contraceptive in Sidama zone is 13% according to report from zonal health department.

### Sample size and sampling technique

The sample size was determined using double population proportion formula using Open Epi version 2.3 with the assumptions of 95% confidence level, power 80%, unexposed-to-exposed ratio of one and proportion of cases among exposed (respectful care by trained service providers) 87% and proportion of cases among controls (respectful care by untrained providers) 80% [27]. Accordingly a sample size of 940 was calculated and proportionally allocated to the health facilities included in the study based on the number of client flow for three months preceding the data collection time. Accordingly, systematic random sampling was used to select every other client who visited family planning units for contraceptive use, and all 66 family planning service providers of the 40 health facilities during data collection period were included. Health facilities (40) were randomly selected from 655 health facilities of Sidama zone.

### Data collection tools and techniques

Data were collected through client exit interview using questionnaire adapted from the Mothers on Respect index (MORi) questionnaire [28]. The MORi questionnaire was developed to assess the client provider interaction and their impact on personal sense of respect. We preferred this tool than other previously used tools because Bohren et al., who developed the most commonly used tool for measuring disrespect and abusive care at health facility, have recommended to develop new validated and reliable tool. Moreover, that tool has a gap in measuring disrespect and abusive care during family planning programme [25]. Accordingly MORi questionnaire is developed through a participatory research process in Canada and USA and the tool was found to be valid and reliable. The tool has 14 items with six point Likert-type scale (ranging from strongly disagree to strongly agree). Very few words like maternity care were modified into family planning context (family planning service) and used directly in this data collection process after translation to the local languages (Amharic and Sidaamu Afoo).

Data were collected through face to face interview with contraceptive user mothers at exit time. To see service provider's characteristics that affect respectful family planning service provision, self-administered questionnaire was used to collect data from service providers using a tool adapted from respectful maternity care training manual developed by the Federal Ministry of Health and Ethiopian Midwives Association [23]. Service providers were given a code without their knowledge when they provide family planning service and after all mothers were interviewed, data were collected from the service providers and linked to maternal data based on the code provided.

**Operational definitions.**   Based on the result of Mother on Respect index (MORi), the range of score is from 14–84 with higher score indicating more respectful care. Accordingly, participant respectful care is classified as: high Respect if the client scores 67–84 (which means they received at least 79.67% services in respectful manner), Moderate Respect if the client scores 50–66 (59.5%– 78.57%), low Respect if the client scores 32–49 (38.1% - 58.33%) and very low Respect if the client scores 14–31 (which means they received respectful care only in 16.67–36.9% case).

In this study, below poverty line is defined as average daily income less than 1.25 dollar (equivalent to 28 Ethiopian birr).

## Data processing and analysis

All quantitative data were checked for completeness, coded and entered in to Epi Data version 3.1 and exported to Stata version 13 for analysis. Frequencies, Mean, standard deviation and proportions were calculated for descriptive purposes and the results were presented using tables and charts. To identify factors associated with the level of respectful family planning service, a partial proportional odds ordinal regression was employed using the *gologit2* command of Stata, since some of the variables violated the proportional odds assumption.

## Ethical approval and consent to participate

Ethical clearance was obtained from the Institutional Review Board of Hawassa University and communicated with Sidama Zone Health department and the selected woredas. A formal letter was obtained from the woredas and communicated with selected health institutions. After the purpose and objective of the study was explained, verbal consent was obtained from each study participant. There was a "Yes/No" question (prompt) on the data collection tool where the data collectors tick whether the selected study participant volunteered or not to participate in the study. If they volunteer to participate, the data collector tick (put" Π") in the box in front of the "Yes" option. In this study, the consent obtained from four participants who were aged less than 18 years were considered valid since all of them were married.

## Results

### Socio-demographic characteristics

A total of 920 individuals from 40 health facilities participated in the study making a response rate of 97.9%. The minimum and maximum ages of the participants were 14 and 46 years, respectively with a mean (±standard deviation [SD]) of 27.19 (± 5.42) years. Among the participants, 905(98.37%) were married, 830(90.22%) were Sidama in Ethnicity, 783 (85.11%) were protestant, 453(49.24%) had elementary school education, 633 (68.8%) identified themselves as housewives and their average annual income ranged from zero to 600,000 birr (equivalent to $21,428.57) with a mean of 26,144.3 birr (equivalent to $933.73) [see Table 1].

**Table 1. Socio-demographic characteristics of study participants, Sidama Zone, Southern Ethiopia, 2018.**

| Variables (n = 920) | Category | Frequency (N = 920) | Percentage |
|---|---|---|---|
| Age | 14–24 | 289 | 31.41 |
| | 25–29 | 322 | 35.00 |
| | 30 and above | 309 | 33.59 |
| Marital status | Single | 8 | 0.87 |
| | Married | 905 | 98.37 |
| | Divorced/widowed | 7 | 0.76 |
| Religion | Protestant | 783 | 85.11 |
| | Orthodox | 53 | 5.76 |
| | Muslim | 33 | 3.59 |
| | Catholic | 24 | 2.61 |
| | Adventist | 27 | 2.93 |
| Ethnicity | Sidama | 830 | 90.22 |
| | Amhara | 45 | 4.89 |
| | Oromo | 13 | 1.41 |
| | Others[a] | 32 | 3.48 |
| Residence | Urban | 273 | 29.67 |
| | Rural | 647 | 70.33 |
| Level of education | no formal education | 271 | 29.46 |
| | Primary (1–8) | 453 | 49.24 |
| | Secondary School (9 -10-) | 152 | 16.52 |
| | Above high school (>10) | 44 | 4.78 |
| Gravidity | Two or less | 506 | 55 |
| | Three and above | 414 | 45 |
| Parity | Two or less | 485 | 52.72 |
| | Three and above | 435 | 47.28 |
| Occupation | House wife | 633 | 68.8 |
| | Employed (government/NGO) | 54 | 5.87 |
| | Private business) | 152 | 16.52 |
| | Student | 67 | 7.28 |
| | Others[a] | 14 | 1.52 |
| Income Category | Under poverty(<1.25$/day) | 465 | 50.54 |
| | Above poverty (>1.25$/day) | 455 | 49.46 |

others[a] Wolayta, Gurage, Tigre, Silte

### Family planning service use

Among the participants, 625 (67.93%) had visited the specific health facility three or more times for family planning service while 137 were first time clients. At the time of the study, 708 (76.96%) of the participants used injectable contraceptive. Waiting time for 766(83.26%) of the study participants was less than 30 minutes. Two hundred eighty five (30.98%) of the participants preferred to get served by males [see Table 2].

**Characteristics of the service providers.** A total of 66 family planning service providers from 40 health facilities participated in this study. The age of the service providers ranged from 22 to 58 years with a mean (±SD) 28.65 (± 5.9) years. Among the service providers who participated in the study, 47 (71.2%) were females, 49 (74.2%) were married, 43 (65.2%) were protestant. Service providers' year of experience ranged from one to 30 years. Their monthly salary ranged from 2,181 birr ($77.89) to 9,000 birr ($321.43) [see Table 3].

Table 2. Family planning service use by the study participants, Sidama zone, Southern Ethiopia, 2018.

| Variables | Category | Frequency (n = 920) | Percentage |
|---|---|---|---|
| Frequency of FP unit Visit | once | 137 | 14.89 |
| | Twice | 158 | 17.17 |
| | Three and above | 625 | 67.93 |
| Method Used | Pills | 247 | 5.11 |
| | Injectable | 708 | 76.96 |
| | Implant | 146 | 15.87 |
| | IUCD | 18 | 1.96 |
| | Condom | 1 | 0.11 |
| Waiting time | ≤30minute | 766 | 83.26 |
| | >30minute | 154 | 16.74 |
| Duration of the procedure | ≤10minute | 757 | 82.28 |
| | >10minute | 163 | 17.72 |
| Number of FP Service providers | One | 615 | 66.85 |
| | Two and above | 305 | 33.15 |
| Sex of FP service providers | Male | 221 | 24.02 |
| | Female | 699 | 75.98 |
| Prefer opposite sex? | No | 635 | 69.02 |
| | Yes | 285 | 30.98 |
| Did FP Provider introduce his name? | No | 745 | 80.98 |
| | Yes | 175 | 19.02 |
| Did FP Provider introduce his role? | No | 778 | 84.57 |
| | Yes | 142 | 15.43 |
| Involved client in decision making | No | 267 | 29.02 |
| | Yes | 653 | 70.98 |
| Client called by her name | No | 358 | 38.91 |
| | Yes | 562 | 61.09 |

*FP, family planning

**Over all respectful family planning service.** In this study almost half of the participants reported that they were highly respected during family planning service utilization and no one reported receiving very low respect. Among the respondents 75 (18.5%) of the family planning service users received low level of respect during family planning service provision, 382 (41.5%) received moderate respect and 463(50.3%) received high level of respect.

## Factors associated with respectful family planning service

In the unadjusted partial proportional odds model, age of the client, place of residence, type of method used (short acting method), clients' level of education, poverty, gravidity, waiting time, duration of procedure, sex of the service provider, training status of service provider on RMC, preference of opposite male service provider by the clients and service providers satisfaction were associated with respectful family planning service at the P-values of < 0.2.

After adjustment using multivariable partial proportional odds model, type of contraceptive used, client's level of education, poverty, preference for male service provider by the clients, service providers' satisfaction and service provider's prior training on respectful care were significantly associated with respectful family planning service provision [**see Table 4**].

**Table 3. Characteristics of service provider on respectful family planning service in Sidama zone, Southern Ethiopia, 2018.**

| Variables | Category | Frequency (N = 920) | Percentage |
|---|---|---|---|
| Age | 22–30 years | 52 | 78.8 |
| | 31 and above | 14 | 21.2 |
| Sex | Male | 19 | 28.8 |
| | Female | 47 | 71.2 |
| Marital status | Single | 17 | 25.8 |
| | Married | 49 | 74.27 |
| Religion | Orthodox | 20 | 30.3 |
| | Protestant | 43 | 65.2 |
| | Muslim | 3 | 4.5 |
| Level of Education | Diploma | 43 | 65.2 |
| | Degree | 23 | 34.8 |
| Profession | Midwifery | 8 | 12.1 |
| | Nurse | 57 | 86.4 |
| | Other | 1 | 1.5 |
| Work experience | ≤2 years | 10 | 15.2 |
| | > 2 years | 56 | 84.8 |
| Ever trained on Respectful care | No | 28 | 42.4 |
| | Yes | 38 | 57.6 |
| Are you Satisfied with your current status? | No | 17 | 25.8 |
| | Yes | 49 | 74.2 |

## Discussion

Our finding showed that 18.5% of the family planning service users received low level of respect during family planning service provision, 41.5% received moderate respect and 50.3%

**Table 4. Ordinal regression table indicating factors associated with respectful family planning service in Sidama zone, Ethiopia, 2018.**

| Variables | Low vs moderate and high | | | Low and moderate vs high | | |
|---|---|---|---|---|---|---|
| | Adjusted odds ratio | P>\|z\| | [95% confidence interval] | | Adjusted odds ratio | P>\|z\| | [95% confidence interval] | |
| Residence (= urban) | 0.774 | 0.110 | .565 | 1.060 | .774 | 0.110 | .565 | 1.060 |
| **Short acting method*** | **0.296** | **0.007** | **.122** | **.722** | **2.106** | **0.000** | **1.421** | **3.122** |
| **Uneducated**** | **0.389** | **0.000** | **.250** | **.606** | **.389** | **0.000** | **.250** | **.606** |
| **Elementary school*** | **0.405** | **0.003** | **.225** | **.729** | .711 | 0.078 | .486 | 1.039 |
| **Under Poverty**** | **0.745** | **0.040** | **.563** | **.986** | **.745** | **0.040** | **.563** | **.986** |
| Gravida3plus | 1.193 | 0.338 | .831 | 1.714 | 1.193 | 0.338 | .831 | 1.714 |
| Age ≤24 | 1.246 | 0.343 | .791 | 1.964 | 1.246 | 0.343 | .791 | 1.964 |
| Age 25–29 | 1.027 | 0.888 | .712 | 1.481 | 1.027 | 0.888 | .712 | 1.481 |
| Wait time >30minute | 1.485 | 0.315 | .687 | 3.207 | **.458** | **0.000** | **.302** | **.693** |
| Duration of procedure >10 minute | .700 | 0.078 | .470 | 1.041 | .700 | 0.078 | .470 | 1.041 |
| Sex of service Provider (= male) | 0.782 | 0.140 | .563 | 1.084 | .782 | 0.140 | .563 | 1.084 |
| **Trained on RMC*** | **8.750** | **0.000** | **4.608** | **16.615** | **3.0322** | **0.000** | **2.245** | **4.096** |
| **Satisfied Providers**** | **1.553** | **0.007** | **1.127** | **2.138** | **1.553** | **0.007** | **1.127** | **2.138** |
| **Client preferred male service provider*** | **1.997** | **0.033** | **1.057** | **3.775** | **.552** | **0.000** | **.400** | **.761** |

*Proportional odds assumption not fulfilled

**Proportional odds assumption fulfilled; RMC, respectful maternity care.

received high level of respect. This is low compared to the current Ethiopian government strategy which estates "zero" tolerance for disrespect and abuse [29].

The odds of moderate or high respectful care for women who were not educated is 61% lesser relative to women who completed at least secondary school, it is 59% lesser for women who attended elementary school compared to those who completed at least secondary school. This is supported by a systematic review conducted in Nigeria which reported that, disrespect and abuse during child birth was more common among women who were uneducated and of low socioeconomic status [16]. Educated women are better aware of their rights reducing the likelihood of being disrespected. Thus, it seems that higher level of education (secondary school and above) protect females from low respect.

The odds of being in the moderate or high category of respectful care relative to being in the low category is 25% lesser for women who were below the poverty line (whose average daily income was less than $1.25) compared to those who were above poverty line. This is supported by studies conducted in Bahir Dar and Addis Ababa where poor women were found to be more abused and disrespected during child birth [17, 30]. Low socio-economic status leads women to seek services in low-quality facilities where women are prone to be disrespected and abused[10].

Though there is no significant difference in the odds of moderate and high respect relative to low respect for women who wait for more than 30 minutes and those who wait less than 30 minutes to get family planning service, the odds of high respect relative to the combined categories of low and moderate respect were 54% lesser for women who wait long (more than 30minutes). Long waiting time may discourage clients and lead them to feel that they are neglected and not respected. In similar studies, long waiting time to contact a service provider was found to be associated with dissatisfaction with family planning service [31, 32].

Women who use short acting contraceptives had a 70% lesser odds of moderate or high respect compared to those who use long acting methods. One of the priority areas of the plan of the Ethiopian government in terms of family planning is to increase the coverage of long acting reversible contraceptives mainly IUCD from 1.1% in 2016 to 8.25% in 2020 and implant from 5% to 18.15% [33]. This strategy may influence service providers to influence clients to take long acting family planning methods. Consequently, clients who prefer short acting methods may be abused and disrespected. On the other hand, the odds of high respect relative to the combined categories low and moderate respect were two times higher for women who use short acting methods compared to those using long acting methods. This could be due to long procedural time for long acting method insertion. In this study, some clients complained that the procedure for IUCD insertion takes around 50 minutes.

The odds of moderate or high respect for clients served by health care workers who were satisfied with their current status were 1.6 times higher relative to those served by health workers who were not satisfied. This is supported with a mapping review and gender analysis study which reports that lack of respect for health care workers and limited training opportunities erode their ability to deliver high quality care [34]. Provider's emotional health has the potential to drive mistreatment and affect women's care [8].

The odds of moderate or high respect are two times higher for women who prefer male service providers. This might occur when those women who prefer male providers are those who need more respect for themselves, need attention and may have good socio-economic status. Secondly, it could be due to cultural expectations that males are more respectful and caring for women than female service providers do.

Women who received service from providers trained on respectful care) had almost nine times higher odds of moderate or high respect compared to those who received service from untrained providers Further, the odds of high respect relative to the combined categories of

low and moderate respect were three times higher for women who received service from trained providers compared to those who received service from untrained providers. Trainings have a positive impact on the quality of counseling provided to clients seeking family planning services, providers' interpersonal skills and overall knowledge [35]. This is supported by a pilot study conducted in Tanzania which showed that training reproductive health care nurses to have a positive impact on promoting respectful care [36].

This study has strengths relative to previous studies. To the authors' knowledge, the present study is the first to assess respectful family planning service provision in Ethiopia (previous studies focus on childbirth). Besides, the study utilized a current validated tool on respectful care which might yield more reliable and valid results. This study has also identified key points affecting client-provider interaction during family planning service provision which may be of importance in informing stakeholders to improve client-provider interaction to ensure respectful family planning service provision. The current tool used to measure respectful service is easily applicable and recommended for use for family planning service, since it measures important areas (autonomy, culture and client provider interaction). This study is not free of limitations. The tool is new and may be difficult for uneducated women to identify the difference in the possible options of the six point scale which may introduce response bias. Secondly, there was skewed distribution of data for type of contraceptive method used and service provided by female service providers.

## Conclusion

The respectful family planning service in the present study is suboptimal, in contrast to the current strategy which allows zero tolerance for disrespect in Ethiopia. Type of contraceptive used, participants' lower level of education, poverty, long waiting time, preference for male service providers, service providers' satisfaction and service providers' prior training on respectful care were significantly associated with respectful family planning service. However, the effect of the identified variables across each level of respectful care is not equal. Almost all factors associated with low level of respectful care are preventable. Strengthening training on compassionate and respectful care is necessary to improve respectful care for family planning clients at all health institutions.

## Supporting information

**S1 File.**
(DOCX)

## Acknowledgments

Our great thank goes to PREPSS at Michigan University, Lee Roosevelt (PhD) and Mr. Habtamu Kebebe (Wollega University) for their comments and assistance in editing the manuscript.

## Author Contributions

**Conceptualization:** Melese Siyoum, Zelalem Tenaw, Abebaw Abeje.

**Data curation:** Melese Siyoum, Abebaw Abeje, Teshome Melese.

**Formal analysis:** Melese Siyoum, Ayalew Astatkie, Zelalem Tenaw.

**Investigation:** Zelalem Tenaw, Abebaw Abeje.

**Methodology:** Melese Siyoum, Ayalew Astatkie, Zelalem Tenaw, Teshome Melese.

**Software:** Melese Siyoum, Ayalew Astatkie, Zelalem Tenaw, Abebaw Abeje, Teshome Melese.

**Supervision:** Melese Siyoum, Ayalew Astatkie, Zelalem Tenaw, Abebaw Abeje, Teshome Melese.

**Validation:** Melese Siyoum, Ayalew Astatkie, Zelalem Tenaw, Abebaw Abeje, Teshome Melese.

**Visualization:** Melese Siyoum, Ayalew Astatkie, Zelalem Tenaw, Abebaw Abeje.

**Writing – original draft:** Melese Siyoum, Ayalew Astatkie, Teshome Melese.

**Writing – review & editing:** Melese Siyoum.

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
