## [Decision Letter · Decision Letter 0]

15 Jun 2020

PONE-D-20-10415

Respectful family planning service provision in Sidama zone, Southern Ethiopia

PLOS ONE

Dear Mr Siyoum,

Thank you for submitting your manuscript to PLOS ONE. After careful consideration, we feel that it has merit but does not fully meet PLOS ONE’s publication criteria as it currently stands. Therefore, we invite you to submit a revised version of the manuscript that addresses the points raised during the review process.

ACADEMIC EDITOR: 

The reviewers noted a couple of points that need to be incorporated in the manuscript. It was also suggested that the manuscript needs a thorough review for its language clarity which I also agree. In this regard, i decided that the manuscript requires a major revision.

We look forward to receiving your revised manuscript.

Kind regards,

Gizachew Tessema, PhD

Academic Editor

PLOS ONE

Journal Requirements:

2. Please provide additional details regarding participant consent. In the ethics statement in the Methods and online submission information, please ensure that you have specified (1) whether consent was informed and (2) what type you obtained (for instance, written or verbal, and if verbal, how it was documented and witnessed). Please also specify if informed consent was obtained from the women seeking care, the care providers, or both. In addition, as your study included minors, state whether you obtained consent from parents or guardians.

3. Please include additional information regarding the survey or questionnaire administered to service providers and ensure that you have provided sufficient details that others could replicate the analyses. For instance, if you developed this questionnaire as part of this study and it is not under a copyright more restrictive than CC-BY, please include a copy, in both the original language and English, as Supporting Information.

4. Your ethics statement must appear in the Methods section of your manuscript. If your ethics statement is written in any section besides the Methods, please move it to the Methods section and delete it from any other section. Please also ensure that your ethics statement is included in your manuscript, as the ethics section of your online submission will not be published alongside your manuscript.

Additional Editor Comments (if provided):

Reviewers' comments:

Reviewer's Responses to Questions

**Comments to the Author**

1. Is the manuscript technically sound, and do the data support the conclusions?

Reviewer #1: Yes

Reviewer #2: Yes

2. Has the statistical analysis been performed appropriately and rigorously? 

Reviewer #1: Yes

Reviewer #2: Yes

3. Have the authors made all data underlying the findings in their manuscript fully available?

Reviewer #1: No

Reviewer #2: No

4. Is the manuscript presented in an intelligible fashion and written in standard English?

Reviewer #1: No

Reviewer #2: No

5. Review Comments to the Author

Reviewer #1: The authors studied an important topic to inform efforts in improving quality of clinical care and provision of client-centered family planning services.

The background presented clearly magnitude of the respectful care or lack of it supported with literature. Overall principles of client-centered care for child birth and family planning are similar. However, there are specificity in defining and addressing individuals’ experiences of care, as well as in the level of client-provider interaction for family planning service. Therefore, rationale for this study can be strengthen with honing in on the family planning service related problems throughout the manuscript.

The sampling and sample size calculation needs further clarity. First, it is not clear how the sample size 940 was determined. It will be helpful if the authors can provide clarity on double population proportion formula used, as it was not clear if the sampling was based on proportions input from two population groups. Second, it was not clear why the 87% proportion was based on a disrespectful care among maternity care clients in Kenya, although the authors presented the proportion for Ethiopia in the background. Third, it will be helpful to clarify how the 40 health facilities were selected out of 130 health center, 522 health post and 3 hospitals in the study area. Forth, it was not clear how many health service providers were interviewed and how they were selected.

The bases for operational definition of the category of high, moderate and low respect should be clearly presented. The referenced article score categorized women who scored in the bottom 10th percentile as those who experienced the least respectful care.

The methodology stated that Mothers on Respect index (MORi) questionnaire adopted with minor changes. However, the authors can clarify if they adopted the binary or the six scale tool. A summary of the changes made to the MORi questionnaire should be presented since the tool was developed for maternity care. Some of the amendments can be presented as the limitation of the original tool, or the necessary adaptation to study site context.

Table 1- it will be helpful to define the cut off point for above and under poverty because there is a wide gap in the presented average income

Table 2- specify if service providers are referring to only family planning service providers in the study facility

Table 3- clarify what ‘satisfied’ provider was referring to

Interpretation of relationship between respectful care with short acting methods as well as with provider gender needs caution with such skewed distribution of the raw data where more than 80% of the clients received short acting methods, 76% of clients were served by female provider and 70% did not prefer opposite sex.

The manuscript will benefit from copy editing.

Reviewer #2: Abstract:

Introduction does not mention family planning and doesn’t specify what aspect of disrespect and abusive care is being referred to (e.g., health care, or more specifically as it relates to family planning service provision).

Results section of abstract is worded awkwardly.

Conclusion: language is not consistent throughout paper. E.g., conclusion refers to disrespect only, not disrespect and abuse.

Background:

There is a large literature on RMC and the authors have done a reasonable job of identifying the key pieces of literature relevant to their study. They are, however, missing some key references that relate RMC specifically to family planning. For example, Harris, Reichenbach and Hardee, “Measuring and Monitoring Quality of Care in Family Planning: Are we Ignoring Negative Experiences?” in Open Access Journal of Contraception, 2016:7, 9-18. This paper reviews the family planning literature using the constructs of D&A from Bowser and Hill.

The background should set the stage better for the unique importance of this study as contributing to the dearth of evidence on the level of respectful care for family planning service provision.

Methods and Materials:

The rationale for the selection of the Mothers on Respect index (MORi) could be explained in greater detail. Why this particular tool? Has it been applied to any non-maternity/delivery examples before this application to family planning? Sharing examples of some of the questions would be useful for many readers. This may also help the reader to understand why there were not more modifications for the family planning context.

The description of the questionnaire administration for the providers is confusing as currently worded.

Results:

Table 2 – some of the variables are not entirely clear and could use rephrasing to be more intuitive for the reader. For example, ‘number of service providers’ is not clear; and ‘involved in decision making’ is unclear – do you mean the client was involved in decision making?

It would be useful to have more description of the finding regarding clients reporting that they were highly respected during family planning service utilization. How was this finding interpreted and what is the definition of being ‘highly respected?’ The operational definition only describes the scale/score and so it is hard to interpret what this means in terms of characteristics of service provision.

What was the justification of setting the level of significance at <0.2? This seems very high.

Discussion:

The discussion section needs a major revision for English and clarity of writing. The findings are certainly interesting but they are hard to tease out given the way the discussion is currently worded.

The discussion section does not describe the implications of the findings. What are the implications for family planning programs?

I was also surprised to not see a more detailed discussion of the degree to which tools developed to measure RMC can be adapted and applied to other health service provision such as family planning. Would the authors further adapt the questions if they were to do the study over again? What would their advice be to other researchers who would like to apply the MORi to family planning? Is there an inherent challenge in using RMC-related frameworks which assess disrespect and abuse at a particular moment in time/single event (labor and delivery) to an ongoing engagement with health providers (provision of most family planning methods)?

The study limitations section needs to be expanded. For example, there are also issues of reporting bias introduced when using self-reported questionnaires such as this.

Conclusion:

The conclusion could be strengthened by expanding the description of the implications of the findings for family planning service provision.

General comment: Careful editing of the manuscript for English is strongly recommended.

6. PLOS authors have the option to publish the peer review history of their article (what does this mean?). If published, this will include your full peer review and any attached files.

Reviewer #1: Yes: Yordanos B Molla

Reviewer #2: No

---

## [Author Response · Author response to Decision Letter 0]

27 Jun 2020

Dear Editor and reviewers, 

thank you very much for your constrictive comments and suggestions. Now we revised the manuscript per suggestions. Here we attached important documents used for this manuscript including tools used for data collection. please see our point by point response for your suggestions. we appreciate your comments a lot.

Regards!

the Authors

Manuscript title: Respectful family planning service provision in Sidama zone, Southern Ethiopia 

Comments from Reviewer #2 Response 

Abstract section:

1. Introduction part lacks information about family planning, Result section is written awkwardly and Conclusion session lacks language consistence 

Thank you a lot for raising these issues.

All comments accepted and corrected. 

The lack of information on family planning in the introduction is because there is lack of literature regarding respectful care among family planning clients. Now, we have described about poor client-provider interaction at family planning units based on available literature.

The whole document has now been thoroughly revised for language. The conclusion sections is also corrected as per the comment. 

4. Background session missed important references 

 Thank you for this comment. We have done the required revisions.

The lack of information on family planning in the background section was because of the lack of literature related to respectful care among family planning service users. Now, we have incorporated evidence from the available literature about family planning in the introduction (see reference 25).

A description of the gap of measuring disrespectful care at family planning unit using tool developed for maternity care is also provided. 

5. Unique Importance of the study need to be strengthened Thank you again 

Importance of the study for family planning programme is well described at the end of background session.

6. Rationale for the selection of the Mothers on Respect index (MORi) could be explained in greater detail Thank you. Now we have described it in more detail. 

This tool was preferred than the others because the previous tools were confirmed to have a gap in assessing respectful care at FP ( see ref 25). 

In addition, the previously used tool (developed by Bohren et al., was not validated). The current tool we used is validated in USA and Canada. 

It can be easily used in family planning context since the focus area was on autonomy (decision making), culture and client-provider interaction. 

7. Questionnaire for service providers were not well described Thank you for pointing this out. Now we have provided a clear description as per the comment.

The questionnaire used to assess service providers’ related factors were adapted from the training manual on respectful maternity care (see ref.23) developed by the Ministry of Health of Ethiopia and Ethiopian Midwives Association. 

8. Table2: some of the variables were not clear and need rephrasing Thank you. Now it is Corrected as: 

a. Service provider =Family planning service providers

b. Involved in decision making = involved client in decision making 

9. Interpretation of the finding needs more description. What is “highly respected mean?” Thank you for pointing this out. Now we have thoroughly revised the description and interpretation of the results in the results and discussion sections. We hope now the interpretations would make more sense.

Further, one advantage of the MORi tool is to show the extent to which the client is respected. Previously respectful care was measured as either 100% or 0% based on whether the client missed one component of care or not. This study shows to what extent the client received respectful care..

10. Level of significance 0.2 seems too high. Thank you for raising the issue. 

In this study Significance level <0.2 was used for binary analysis, it is not for the final analysis (multivariate). 

11. Discussion section doesn’t describe implication of the finding Thank you. Revised accordingly.

The implication of the finding is included at the end of discussion and conclusion sections. 

12. Discussion section needs major revision for English and write up Thank you for pointing this out. Now we have thoroughly revised the discussion section. 

The overall write up and language was edited by a pre-publication support service (PREPSS) from Michigan University. 

13. Limitation section and conclusion part need to be more elaborated Thank you for this comment. Done accordingly.

Elaborated by including the implication of the findings. 

Comments from reviewer #1 Response 

Background 

1. Rationale of this study need to strengthen by honing on family planning related problems. Thank you for raising this very important issue. Now we tried to strengthen it by incorporating important points. 

Poor client-provider interaction at family planning unit was described. Additional references were added. The gap of measuring disrespectful care at family planning unit using tool developed for maternity care was clearly stated. Moreover, lack of evidence on family planning is incorporated (see ref 25). 

Importance of the study for family planning programme is well described at the end of back ground session.

2. Clarify whether the binary or six scale questionnaire was adopted from MORi tool? Thank you again. 

It was described under methodology section that we used the six scale tool because it was confirmed that the six scale tool well measures the level of respectful service provided. 

3. Describe the changes(modifications) made to the questionnaire Thank you very much. 

The only modification made to the tool was that we modified “maternity care” in to “family planning service”. It can be easily used in family planning context since the focus area was on autonomy (decision making), culture and client-provider interaction. 

4. The base for operational definition is not clear Thank you again. 

We attached here a one page of the MORi tool as supporting file. It clearly state how to classify the level of respectful care (see S1). 

5. Sample size determination is not clear as whether the proportions were taken from two populations. 

 Thank you for raising this issue, 

Yes the proportions were taken from two populations. Respectful service provision was 87% and 80% by trained and untrained providers in Kenya. Now we have corrected the description as per the comment. 

6. Why the proportions were taken from study in Kenya while proportion were reported from Ethiopia. Thank you for this question. 

This study was needed for intervention to improve respectful family planning service by Center of International Reproductive Health Training (CIRHT) Ethiopian branch. So we preferred to use proportions from interventional studies. Currently different interventions are being implemented in this study area and its impact will be assessed near the future. 

7. It is not clear how 40 health facilities were selected Thank you for this comment. Now we have clearly stated in the methodology section how the 40 health facilities were selected.

These 40 health facilities were selected randomly from all health facilities in the zone. 

8. Number of service providers participated Now it is clearly stated

All 66 family planning service providers during data collection period were included. 

9. Table1: cut off point for under-poverty is needed Thank you for raising this important question. 

Now we described it within the table and under operational definition sections. 

Income was categorized as under-poverty if their average daily income is equivalent to <1.25$. 

10. Table2: service providers were not specifically described Thank you. Now it is corrected 

Service provider means Family planning service providers by the time of data collection. 

11. Table3: clarify “satisfied provider”. Thank you. Now it is clearly described. 

It is service provider’s perceived satisfaction with their current status. 

12. Interpretation for relationship between respectful care with short acting methods as well as with provider gender needs caution with such skewed distribution of the raw data Thank you very much for raising this important issue. 

As you see it from the regression table, the finding is significant with acceptable confidence interval. However, the skewed distribution of the raw data is now described as a limitation. 

13. The manuscript need edition for language clarity. Thank you for pointing this out. Now we have thoroughly revised the discussion section. 

The overall write up and language was edited by a pre-publication support service (PREPSS) from Michigan University.

---

## [Decision Letter · Decision Letter 1]

21 Aug 2020

Respectful family planning service provision in Sidama zone, Southern Ethiopia

PONE-D-20-10415R1

Dear Mr Siyoum,

We’re pleased to inform you that your manuscript has been judged scientifically suitable for publication and will be formally accepted for publication once it meets all outstanding technical requirements.

Kind regards,

Gizachew Tessema, PhD

Academic Editor

PLOS ONE

Additional Editor Comments (optional):

Reviewers' comments:

Reviewer's Responses to Questions

**Comments to the Author**

1. If the authors have adequately addressed your comments raised in a previous round of review and you feel that this manuscript is now acceptable for publication, you may indicate that here to bypass the “Comments to the Author” section, enter your conflict of interest statement in the “Confidential to Editor” section, and submit your "Accept" recommendation.

Reviewer #1: All comments have been addressed

2. Is the manuscript technically sound, and do the data support the conclusions?

Reviewer #1: Yes

3. Has the statistical analysis been performed appropriately and rigorously? 

Reviewer #1: Yes

4. Have the authors made all data underlying the findings in their manuscript fully available?

Reviewer #1: Yes

5. Is the manuscript presented in an intelligible fashion and written in standard English?

Reviewer #1: Yes

6. Review Comments to the Author

Reviewer #1: The manuscript has improved significantly. Thank you for providing an insight into critical issues to improve quality of clinical care and provision of client-centered family planning services.

7. PLOS authors have the option to publish the peer review history of their article (what does this mean?). If published, this will include your full peer review and any attached files.

Reviewer #1: **Yes: **Yordanos B Molla

---

## [Editor Report · Acceptance letter]

27 Aug 2020

PONE-D-20-10415R1 

Respectful family planning service provision in Sidama zone, Southern Ethiopia 

Dear Dr. Siyoum:

I'm pleased to inform you that your manuscript has been deemed suitable for publication in PLOS ONE. Congratulations! Your manuscript is now with our production department. 

Kind regards, 

on behalf of

Dr. Gizachew Tessema 

Academic Editor

PLOS ONE